# Automated Identification of Mineral Types and Grain Size Using Hyperspectral Imaging and Deep Learning for Mineral Processing

**Natsuo Okada [1],\*, Yohei Maekawa [2], Narihiro Owada [3], Kazutoshi Haga [1], Atsushi Shibayama [1] and Youhei Kawamura [1]**

[1] Graduate School of International Resource Sciences, Akita University, 1-1 Tegata Gakuen-machi, Akita 010-8502, Japan; khaga@gipc.akita-u.ac.jp (K.H.); sibayama@gipc.akita-u.ac.jp (A.S.); y.kawamura@gipc.akita-u.ac.jp (Y.K.)

[2] Department of Transdisciplinary Science and Engineering, School of Environment and Society, Tokyo Institute of Technology, I4-21, 2-12-1 Ookayama, Meguro-ku, Tokyo 152-8552, Japan; maekawa.y.ag@m.titech.ac.jp

[3] Faculty of International Resource Sciences, Akita University, 1-1 Tegata Gakuen-machi, Akita 010-8502, Japan; s1017310@s.akita-u.ac.jp

\* Correspondence: natsu0807chi@gmail.com; Tel.: +81-809-525-1518

**Abstract:** In mining operations, an ore is separated into its constituents through mineral processing methods, such as flotation. Identifying the type of minerals contained in the ore in advance aids greatly in performing faster and more efficient mineral processing. The human eye can recognize visual information in three wavelength regions: red, green, and blue. With hyperspectral imaging, high resolution spectral data that contains information from the visible light wavelength region to the near infrared region can be obtained. Using deep learning, the features of the hyperspectral data can be extracted and learned, and the spectral pattern that is unique to each mineral can be identified and analyzed. In this paper, we propose an automatic mineral identification system that can identify mineral types before the mineral processing stage by combining hyperspectral imaging and deep learning. By using this technique, it is possible to quickly identify the types of minerals contained in rocks using a non-destructive method. As a result of experimentation, the identification accuracy of the minerals that underwent deep learning on the red, green, and blue (RGB) image of the mineral was approximately 30%, while the result of the hyperspectral data analysis using deep learning identified the mineral species with a high accuracy of over 90%.

**Keywords:** mineral processing; mineral identification; CNN; machine learning

---

## 1. Introduction

### 1.1. Background

The recent increase in resource demand in emerging economies has caused the rise of resource nationalism in resource-rich countries in the global competition for mineral resources. Since the average grade of ore deposits decreases over time as these are non-renewable resources, it is necessary to increase the efficiency of resource development. Rock classification plays a crucial role in this [1–3]. Developing efficient mineral processing technology for low-grade minerals will be important. We expect that more tailings (low-grade minerals) will be produced in the coming years due to the increased exploration of low-grade minerals [4,5]. After mining the ore, if the appropriate mineral processing method can quickly be selected for each mineral species and grain sizes, this will lead to a more efficient operation.

A schematic diagram of this process is shown in Figure 1. The implementation stage envisioned by the system is the stage after mining, after sorting of the rough grain size and color sorting of minerals and before beneficiation is carried out. The main purpose of the system is to assist in the implementation of more useful beneficiation by identifying the mineral species to be beneficiated and the grain size of the minerals. This identification plays a complementary role in assisting the beneficiation process. To this end, it provides economic, production, and environmental data on which to base decisions for mining and process planning. After identification of the minerals by deep learning, depending on the size of the grains, the separation of the minerals is performed using air shots and rail switching equipment to separate high grade and low grade minerals.

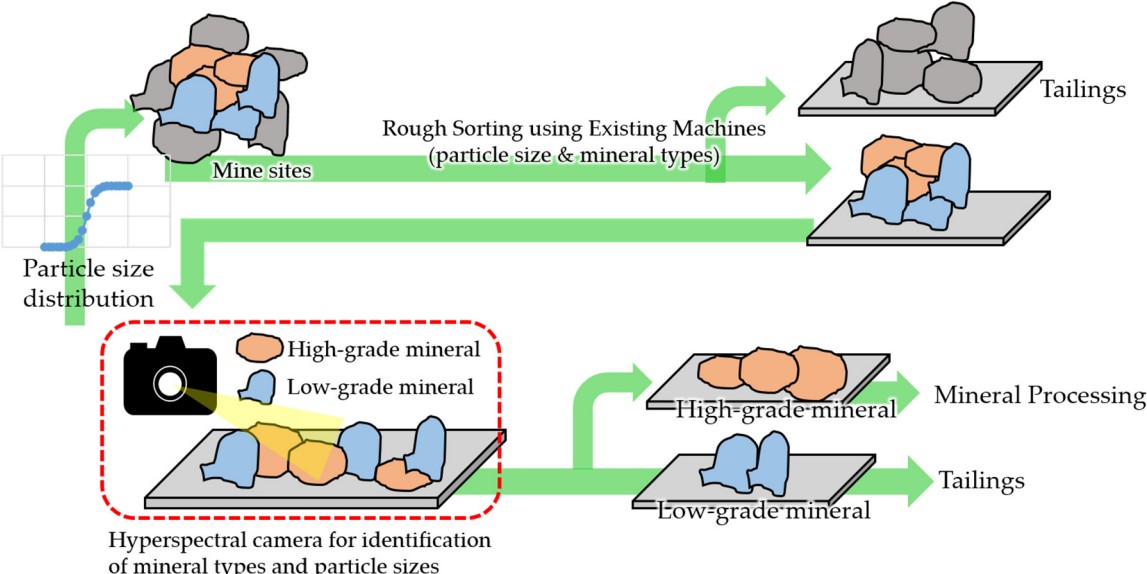

**Figure 1.** Schematic diagram of mineral processing using hyperspectral imaging and deep learning classification.

This system is designed to perform a more detailed size determination after the ore has been mined and roughly sorted by existing sorters. As a result, it is possible to perform a more detailed size determination on ores that have already been sorted into several types, making it possible to measure data that complements the particle size distribution measurement. In addition, mining of hematite, for example, is carried out by blasting with explosives; however, if the grain size of the mineral is too fine during blasting, it can lead to a dilution of the mineral grade. This system optimizes blasting by feeding back the grain size data of the ore after mining to the blasting operation.

For identification of minerals, the industry standard utilizes analysis equipment, such as X-ray diffraction (XRD) and inductively coupled plasma optical emission spectroscopy (ICP–OES); however, these are laboratory processes, and the amount of analyzed samples is relatively small compared to the run-of-mine as the process requires significant time and resources to perform [6]. With regard to the identification of mineral species in beneficiation, techniques and approaches have been developed and established, particularly in the fields of geometallurgy and process mineralogy [7–12].

Geometallurgy promotes sustainable development when all stages of extraction are performed in an optimal manner from a technical, environmental, and social perspective [13]. The discipline of process mineralogy developed through the recognition that metallurgical flowsheets could be optimized by thorough characterization of the precursor ore mineralogy, mineral associations, grain size and textures [14]. Geometallurgy and process mineralogy as multi-disciplinary fields have been applied at various levels in different operations [15], and these have evolved to incorporate machine learning in an attempt to build more optimal beneficiation processes.

On the background of these recent efforts, we developed a new approach to the development of block models and mineral evaluation for geometallurgy and process mineralogy by combining hyperspectral imaging [16,17] and deep learning for pre-fertilized ores. We named the system "pre-mineral processing". This system is implemented for the ores after mining and for the ores before the beneficiation process, and the obtained data, such as mineral species and grain size, are provided as supplementary data for the later beneficiation process.

Deep learning is a machine learning model that is significantly higher in performance than other conventional models in that it can automatically extract the featured amount of data, and has achieved results in various fields [18–20]. In the field of mineral processing, the application of deep learning has not been fully explored yet, and this research aims to contribute to its advancement in the field of mineral processing. Another method for mineral identification is for a geologist or a person with equivalent expertise to visually inspect the samples. While this method is fairly accurate with respect to the capability of the expert albeit with some bias, it is not practically possible for them to observe a large amount of samples. In this case, the proposed system might be preferable. The information that can be obtained with the naked eye is a red, green, and blue (RGB) image consisting of three wavelength bands of red, blue, and green, and it is thought that deep learning of these data will enable minerals to be discriminated as an expert does.

In this research, spectral data of the wavelength region from visible light region to a part of near infrared region (400–1000 nm) was obtained and analyzed. Normally, humans can recognize three wavelength regions associated with red, blue, and green; however, it is possible to obtain 66 high resolution spectral bands as data using hyperspectral imaging. Since the acquired hyperspectral data includes tens of thousands of relatively large data, this was processed using deep learning that can analyze large amounts of data. For this purpose, the study used a convolutional neural network (CNN). A CNN can automatically extract the features of the input image data, and in this study, the features, such as wavelength peaks and slopes peculiar to each mineral of the input spectral data were automatically extracted and learned [17,21]. For the purpose of identifying mineral types and grain size, as experimental samples, chalcopyrite, galena, and three different particle sizes of hematite were selected for the identification of rock types for mineral processing and hyperspectral data of those five types of specimens were obtained and analyzed using CNN to determine if the CNN could perform an acceptable level of mineral classification.

These five minerals were selected for the experiment for two reasons. First, chalcopyrite and galena are sulfide minerals, and hematite is an oxide mineral, and this system enables us to classify sulfide and oxide minerals. Secondly, this is because the grain size can be controlled by discriminating the grain size in the process, which makes it possible to improve the efficiency of mineral processing. At an actual operation site, there are only a few types of ores to be sorted [1,3,4,22,23]. Therefore, it was assumed that the classification performance is sufficient for only a few types of minerals to be classified at an operation site. This study focused on the classification of sulfide and oxide minerals and the identification of grain size, which are important processes in mineral separation.

First, an experiment was conducted to identify only the mineral species, then the three hematite species with different grain sizes, and finally, to identify all of them together. The hematite at the Waga Sennin deposit in Iwate Prefecture, the minimal hematite at the Ani deposit in Akita Prefecture, the galena at the Agenosawa Mine in Akita Prefecture, and the chalcopyrite at the Tada Mine in Hyogo Prefecture. The system using CNN is more versatile not only for identifying mineral types but also for classifying the particle sizes of hematite.

## 1.2. Technics

Experts, such as geologists, identify mineral species visually by observing the external mineral characteristics, such as the color, streak, or reflectance of minerals, that are unique to certain mineral types. In deep learning, the data of minerals is randomly learned to recognize the patterns that define the mineral species, and the classification is performed by extracting the unique features that define the

mineral species in a similar manner as a human expert would. By adopting deep learning, it is possible to analyze visual data at a much higher speed than humans. Visual identification is discrimination based on RGB images; however, when experts identify mineral types not only do they see visual data but they also consider characteristics, such as the specific gravity or hardness of the minerals.

Our study, therefore, adopted hyperspectral imaging to compensate and allow for accurate mineral identification using only visual data. As such, data processing was performed using machine learning, which is capable of handling large amounts of data. Machine learning that automatically learns patterns from a large amount of data is classified into two models, supervised learning and non-supervised learning [24]. Supervised learning is a learning method where the input data are labeled and on the other hand, non-supervised learning is a method without labeling. In this study, the hyperspectral data of the obtained ore were assigned the ground truth label of the ore type and used for supervised learning. Unsupervised learning was not used in this study due to the lack of classification based on mineral species and the hyperspectral data of the minerals being acquired as the labels of the data were known at the time they were acquired.

In supervised learning, CNN was adopted, which is a deep learning method capable of automatically extracting the features of input data. From the input mineral spectrum, CNN automatically recognized, extracted, and learned the spectral shapes that are considered to be mineral-specific, such as the light reflection intensity, slope, and peak. Although CNNs are affected by noise, the effect becomes negligible as the amount of data increases. In this research, we analyzed the mineral's spectral tendencies, such as peaks or intensity and slopes of spectral data using a CNN that specializes in image processing among deep learning. CNNs are neural networks that perform learning by repeating the operation and emphasizing the characteristics of the input data. We performed an operation called convolution on the input RGB image, shown in Figure 2, and on the input hyperspectral data, shown in Figure 3. Here, the convolved output data is represented by Equation (1) where x is the image, m × n is the kernel size, w is weight, and b is for bias [17,21].

$$a_{ij} = \sum_{s=0}^{m-1} \sum_{t=0}^{n-1} w_{st} x_{(i+s)(j+t)} + b \tag{1}$$

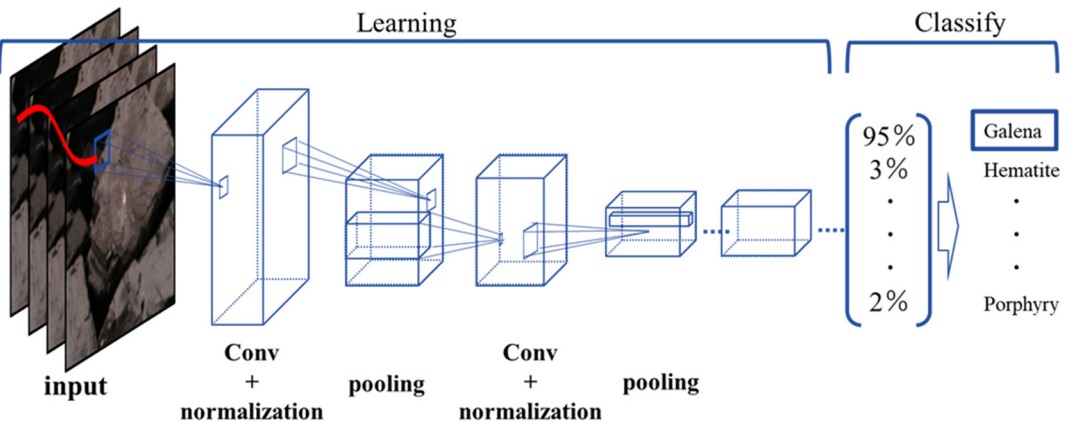

**Figure 2.** Structure of the convolutional neural network (CNN) for red, green, and blue (RGB) images.

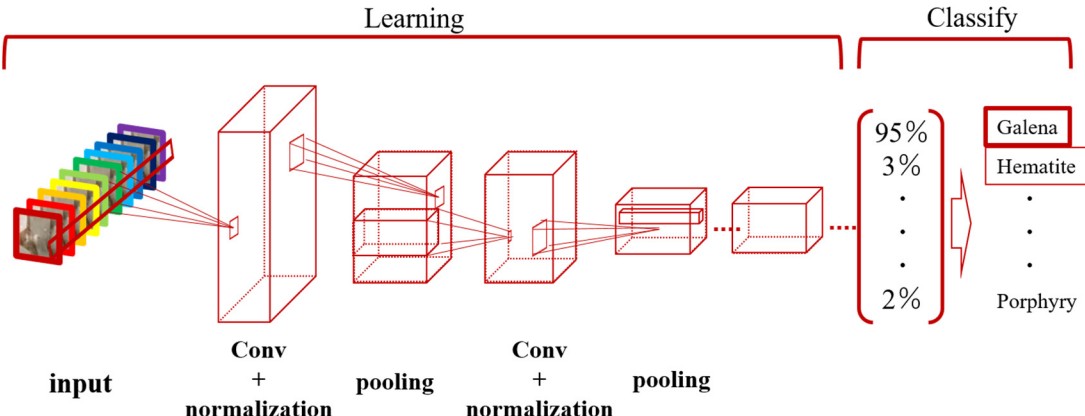

**Figure 3.** Structure of the CNN for hyperspectral data.

The output data a is collated with the correct answer label, and, if there is an error, the weight w is updated, and learning is performed from the beginning. CNN has several convolution layers as it calculates convolution multiple times. Images with emphasized characteristics pass on to the next convolution layer and, as the layers go deeper, the images are emphasized more [22]. The spectral data for learning are 1-D data; however, we considered these data as pseudo 2-D data and input them to the CNN. We verified the effectiveness of hyperspectral data for mineral identification by comparing with RGB images. For RGB images, we adopted transfer learning using GoogLeNet [25] that was pre-trained with other data sets to save on learning costs and increase the efficiency. GoogLeNet network was used with a depth of 22 layers, which has already been trained for 1000 kinds of images, and we retrained part of the network by modifying the layers for the dataset used in this study. The architecture of GoogLeNet is shown in Figure 4 [25].

For the hyperspectral data, we constructed a neural network, consisting of a convolutional layer, normalization layer, pooling layer, and softmax layer, from scratch as there was no pre-trained network for hyperspectral data, as shown in Table 1. Both networks have the same structure in terms of both having a convolution layer to extract characteristics, a ReLu layer for use as an activation function, and an output layer for the output results. The neural network was optimized by matching the predict labels output from the CNNs with the input ground truth labels and feeding this back to the network using validation data. Ground truth labels indicate the name of the mineral type and predict labels indicate the name of the mineral predicted by the CNN.

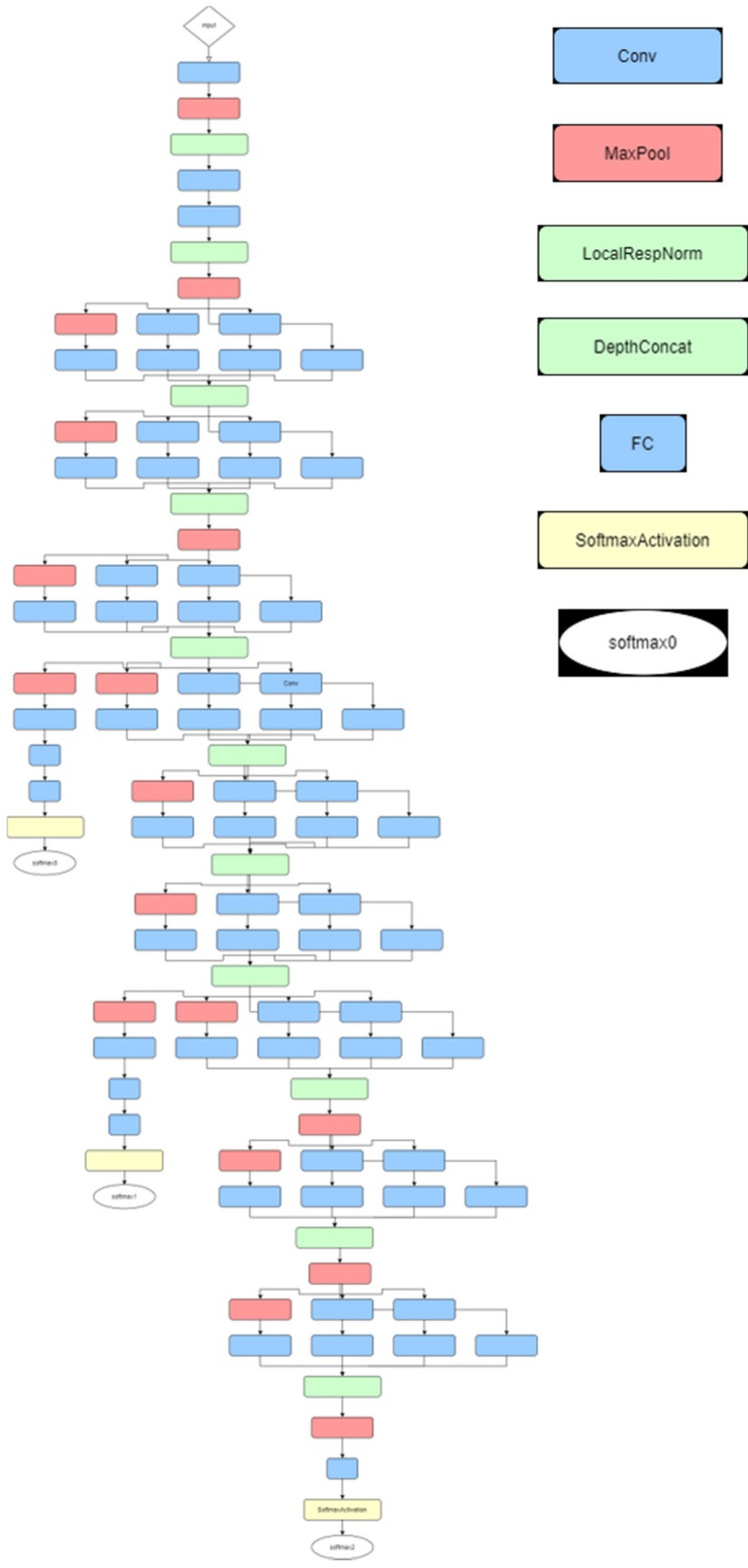

**Figure 4.** An architecture of GoogLeNet.

**Table 1.** Layers of CNN use for hyperspectral data analysis.

| Layers | Layer Size | Stride | Padding Size | Layers | Layer Size | Stride | Padding Size |
|---|---|---|---|---|---|---|---|
| ImageInputLayer | [1, 204] | - | - | Convolution2dLayer | [1, 3] | 1 | [0, 1] |
| Convolution2dLayer | [1, 3] | 1 | [0, 1] | Convolution2dLayer | [1, 3] | 1 | [0, 1] |
| Convolution2dLayer | [1, 3] | 1 | [0, 1] | BatchNormalizationLayer | - | - | - |
| BatchNormalizationLayer | - | - | - | ReluLayer | - | - | - |
| ReluLayer | - | - | - | MaxPooling2dLayer | [1, 2] | [1, 2] | - |
| MaxPooling2dLayer | [1, 2] | [0, 1] | | FullyConnectedLayer | - | - | - |
| Convolution2dLayer | [1, 3] | 1 | [0, 1] | FullyConnectedLayer | - | - | - |
| Convolution2dLayer | [1, 3] | 1 | [0, 1] | FullyConnectedLayer | - | - | - |
| BatchNormalizationLayer | - | - | - | SoftmaxLayer | - | - | - |
| ReluLayer | - | - | - | ClassificationLayer | - | - | - |
| MaxPooling2dLayer | [1, 2] | [1, 2] | - | | | | |

## 2. Materials and Methods

### 2.1. Capturing RGB Images

Figure 5 shows one example RGB images of each (a) hematite with large particles (0.5–1.0 cm), (b) hematite with small particles (<100 µm), (c) hematite with very small particles (<20 µm), (d) galena, and (e) chalcopyrite. In this study, to compare with the hyperspectral data, we examined whether minerals could be identified using RGB images of minerals acquired by deep learning.

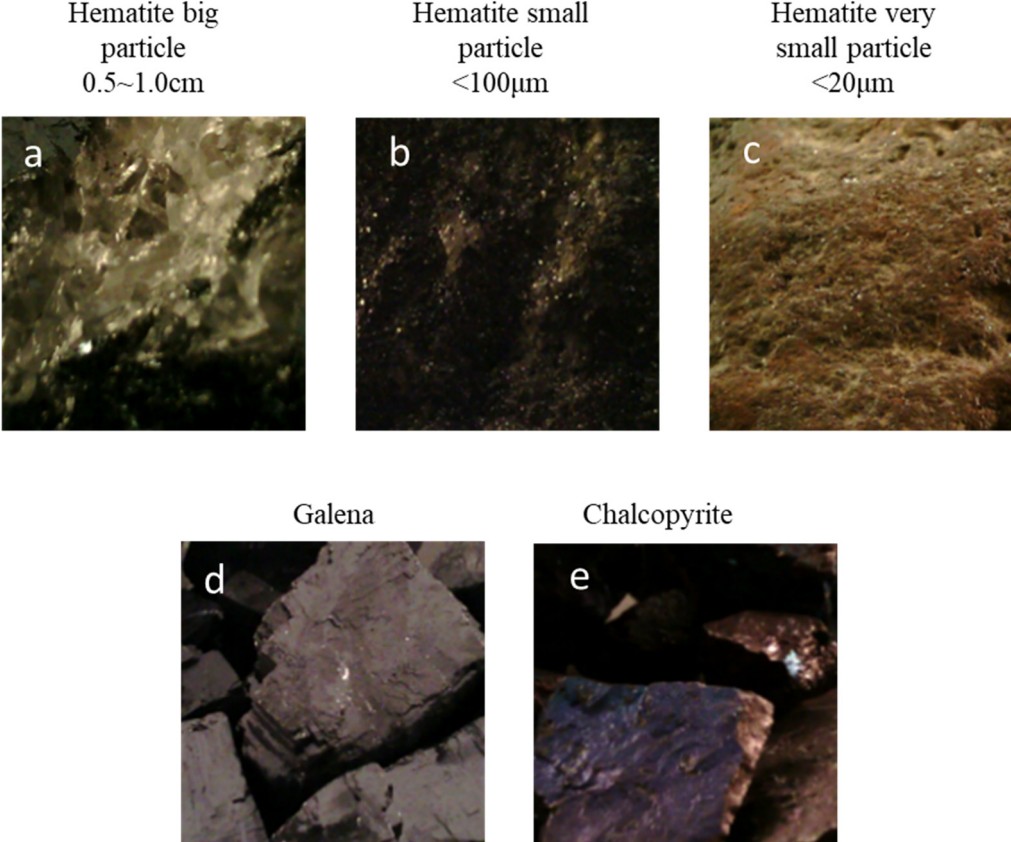

**Figure 5.** Five types of minerals for classification: (**a**) hematite large particles, (**b**) hematite small particles, (**c**) hematite very small particles, (**d**) galena, and (**e**) chalcopyrite.

### 2.2. Capturing Hyperspectral Image

A hyperspectral camera is a special camera that can take a photograph by spectrally splitting the light for each wavelength. Specim IQ produced by Spectral Imaging Co. was used in this study and can split the wavelength of light from 400 nm to 1000 nm, which is part of the visible light region to the near infrared region, into 204 wavelength bands. The wavelengths that can be recognized with the naked eye are the three wavelength bands of red, blue, and green, and it is possible to obtain high resolution wavelengths by using a hyperspectral camera. Since the identification of minerals by experts is possible with the naked eye, it is considered that the ore has unique characteristics, such as the reflection of the light in the hyperspectral data that is more detailed than the information obtained with the naked eye. In this research, we analyzed those spectrum data of five types of minerals using deep learning as shown in Figure 5.

## 2.3. Experiment

As shown in Figure 6a,b, a target mineral was illuminated with halogen lights from three directions in a dark room to serve as a light source. This can illuminate all bands of the spectrum from 400 nm to 1000 nm using halogen light but not with fluorescent light and LED. The halogen light filament is heated over time and its wavelength peak transitions to the higher side; however, this effect was not seen in the wavelength data acquired after several hours of use from the previous experiment, and thus we ignored the effect. In a standard camera, the light is divided into three equal parts, red, blue, and green, and the light reflected from these is acquired. In a hyperspectral camera, the light is divided into 204 equal parts of the same intensity; therefore, the light must be more intense than in a standard camera. Increasing the exposure time can compensate for this, but the simplest way is to increase the intensity of the illumination.

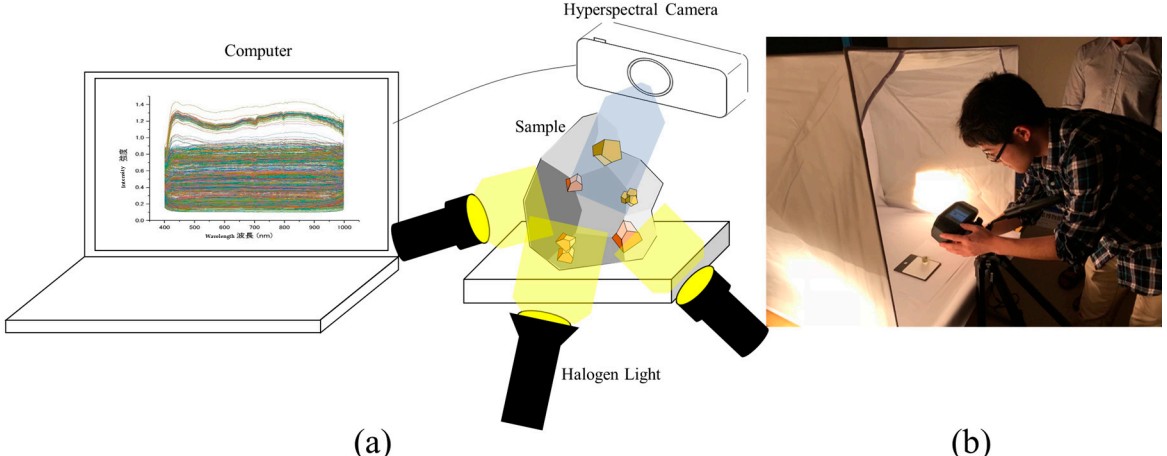

(a)                                                                          (b)

**Figure 6.** Arrangement of the experiment: (**a**) schematic diagram of experiment, and (**b**) photo of the set-up of experiment.

To further equalize the intensity of the light obtained in each region in the hyperspectral data acquired, the number of lights and the angle of each light source were arranged so that no shadows were created as shown in Figure 6. A single image capturing time by the hyperspectral camera is completed within about 1 min, and the imaged data can be shown as wavelength spectrum data on a computer as shown in Figure 6. For the minerals, the data were acquired with a hyperspectral camera with the flat surface facing out to avoid shadows caused by the light source. Although the surface roughness of each mineral varied, shadows were eliminated as much as possible by the placement of the light source. Therefore, the quality of the acquired data ensures the robustness of the deep learning model. At the actual mine site, the ore can be sorted by acquiring and analyzing hyperspectral data of the ore prior to beneficiation. In this experiment, we strictly adjusted the light intensity to obtain accurate data; however, there was no need to control the light intensity because the exposure time is adjusted on the camera side. In addition, because deep learning determines the mineral species for each acquired hyperspectral image, even if some data are inadequate, if the majority of the data is normal, this does not have a significant impact on the overall mineral determination results.

The data acquired by the hyperspectral camera is called a data cube, and has a data storage format of 512-pixels in the vertical direction, 512-pixels in the horizontal direction, and wavelength data corresponding to the depth of 204-pixels. In deep learning, the accuracy of learning increases as the number of data increases. In this study, to increase the data set for deep learning, we divided the vertical and horizontal pixels into 16 equal parts, and prepared 256 data cubes of 32-pixels each in the vertical and horizontal directions and 204-pixels in depth for each image capture. One of the characteristics of hyperspectral data is that the wavelength range acquired is wider than that of RGB

images. Therefore, instead of slicing the hyperspectral data and analyzing it by deep learning, we split the hyperspectral data in the direction of the wavelengths to analyze the wavelength data.

We divided the hyperspectral data by 16 to reduce the computational requirements needed to classify the spectral anomalies, and to standardize the data so that noise or abnormalities in individual pixels did not affect the overall capability of the CNN. Thereby, the number of pixels was chosen as it respected the significance of every anomaly, without averaging so much that the anomalies from the $512 \times 512$ images were insignificant. Then, the 16 divided data were averaged so that each vertical and horizontal pixel was 1-pixel, and processed into data of one vertical and one horizontal pixel and 204-pixels deep as shown Figure 7.

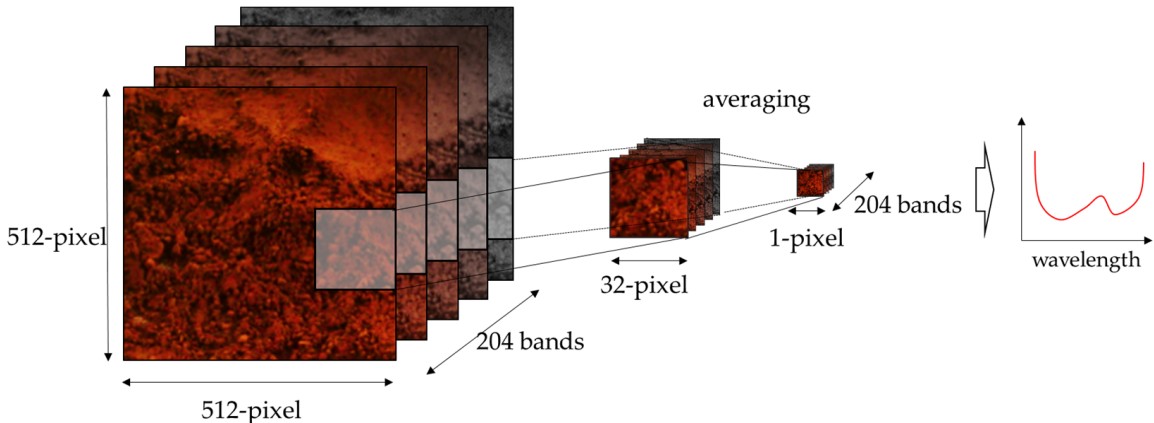

**Figure 7.** Processing of the hyperspectral data for deep learning.

### 2.4. Obtained Images

Figure 8 shows the spectrum data obtained using the hyperspectral camera. The horizontal axis represents the wavelength (nm), the vertical axis represents the reflection intensity, and each colorful line represents the obtained spectrum data. Hyperspectral imaging is more informative than RGB images considering that a hyperspectral image contains 66 times the wavelength region compared with RGB images. We show the five types of minerals in Figure 5. As it is difficult to manually extract the characteristic peaks and slopes that determine the mineral species from the spectral shapes of these minerals, we extracted these features automatically using deep learning. In addition, the data were not normalized to the original data because the data were normalized in the image input layer. Hyperspectral data includes as many as 204 wavelength bands, but deep learning enables processing of these large amounts of data. Figure 9 shows the difference of the spectrum bands between RGB images and hyperspectral data. The figure on the left (a) is an overview of the RGB image spectrum, and the figure on the right (b) is an overview of the hyperspectral data.

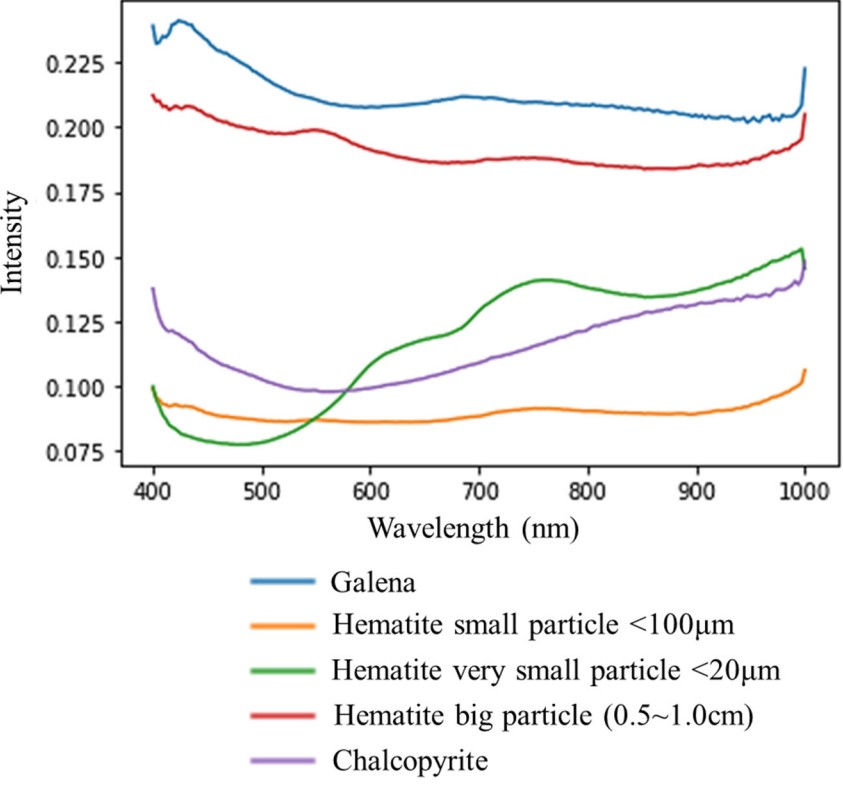

**Figure 8.** Averaged obtained hyperspectral data of each mineral.

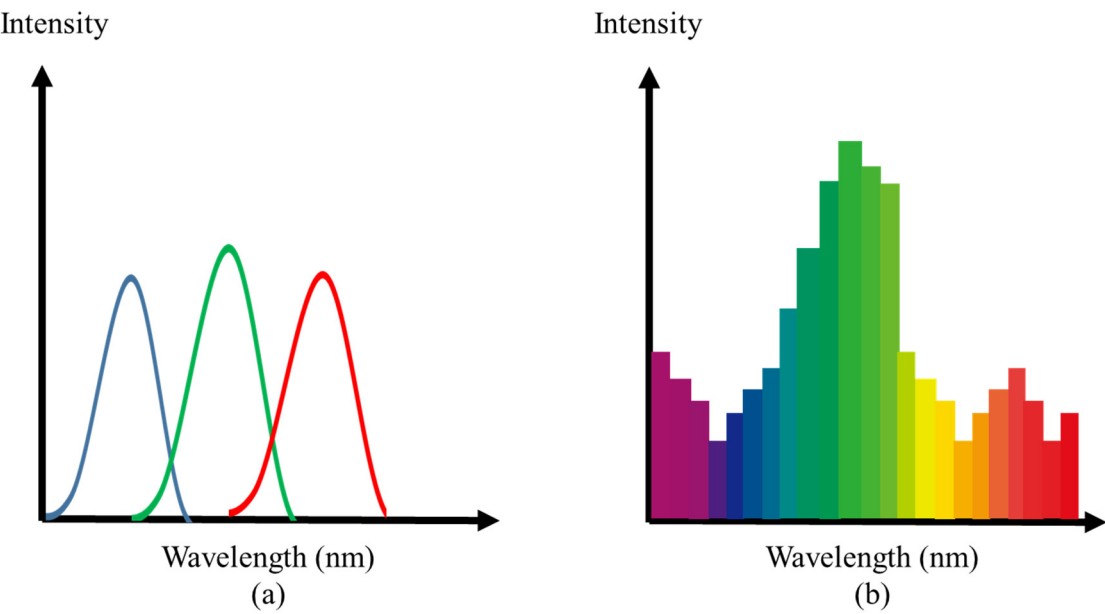

**Figure 9.** The difference between (**a**) an RGB image and (**b**) hyperspectral imaging.

## 3. Results

### 3.1. Deep Learning Analysis for RGB Images

When capturing images with a hyperspectral camera, an RGB image, as shown in Figure 5, is also acquired separately from the image cube. The number of data was increased by dividing the acquired RGB image. To compare the mineral identification accuracy between hyperspectral data and RGB images, deep learning was performed for the RGB images of the five types of minerals. To accelerate the

processing speed of the data, we constructed the network by selecting transfer learning, which performs learning with a pre-trained network. As the network weights were already adjusted in transfer learning, only the last learning layer and classification layer of GoogLeNet were replaced with the learning layer for the RGB image data set of five types of minerals, and retraining was performed. The learning data set is shown in Table 2. The data set consisted of training data, validation data, and test data. The training data was used for training the network, the validation data was used for feedback of the output result of training, and the test data was used for testing the trained model. The size of the RGB images was $16 \times 16 \times 3$ pixels.

**Table 2.** Data set of the RGB images.

| RGB Images $16 \times 16 \times 3$ Pixels | Training Data | Validation Data | Test Data | Total |
|---|---|---|---|---|
| Galena | 770 | 96 | 97 | 963 |
| Chalcopyrite | 770 | 97 | 97 | 994 |
| Hematite large particles | 765 | 95 | 96 | 956 |
| Hematite small particles | 806 | 101 | 101 | 1008 |
| Hematite very small particles | 819 | 103 | 102 | 1024 |

The results of the analysis are shown in Figure 10a for the learning curve and Figure 10b for the loss function. In Figure 10a, the horizontal axis shows the iteration and the vertical axis shows the accuracy of classification. In Figure 10b, the horizontal axis shows the iteration and the vertical axis shows the loss of classification. The learning accuracy was obtained by using the number of classified data as the denominator and the number of correct data as the numerator. Looking at the learning curve of Figure 10a, there was no increase in the accuracy of classification for the training data indicated by the blue line from the beginning to the end of the iteration. The training accuracy was lower than the validation accuracy and the training loss was higher than the validation loss because GoogLeNet included a dropout method that disabled the hidden layer of neurons during training with a fixed probability.

At the time of validation, the training accuracy was lower than the validation accuracy because the loss was calculated in a more robust network that did not include the dropout. On the other hand, at validation, the training accuracy was lower than the validation accuracy because the loss was calculated in a highly robust network that did not include the dropout. This shows that it is difficult to identify minerals using only RGB images. Even when a specialist identifies a mineral, not only the information on the surface of the mineral but also a combination of several methods, such as scratching the surface of the mineral to see the striation color, seeing the transparency through light, are used. It may be difficult to identify minerals only by the RGB information. The results of the final iteration of learning accuracy of Figure 10 are shown in Table 3. The accuracy was 39.52% for the classification of five types of minerals from RGB images.

**Table 3.** Result of deep learning for RGB images for the learning data.

| | Five Types of Minerals |
|---|---|
| Final accuracy | 39.52% |

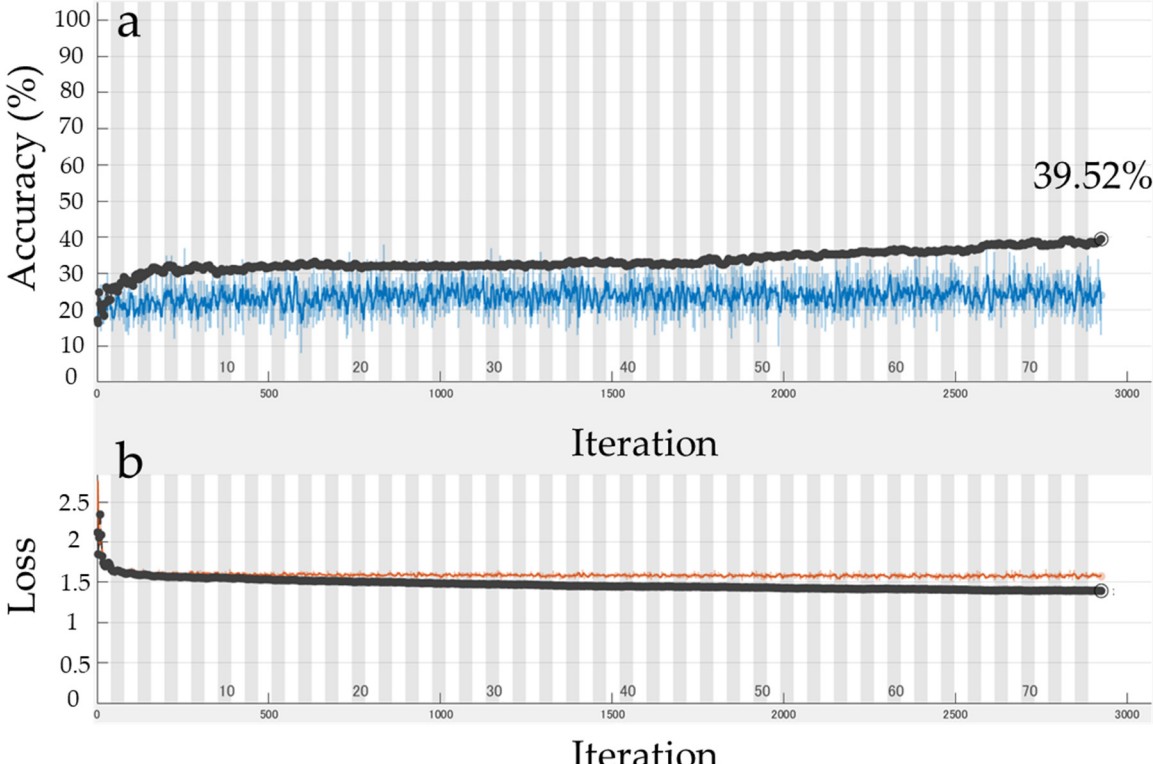

**Figure 10.** (**a**) Learning curve of the RGB images using deep learning. (The horizontal axis shows the iteration and the vertical axis shows the accuracy of classification. The blue line shows the accuracy for the learning data set and the black dots represent the accuracy for the verification data set.) (**b**) The loss function of RGB images using deep learning. (The horizontal axis shows the iteration and the vertical axis shows the loss of classification. The orange line shows the loss for the learning data set and the black dots represent the loss for the verification data set.).

In addition, Figure 11 shows the confusion matrix of the learning results of the classification of the five types of minerals, in which the performance of the classifier was evaluated using test data that was not used for learning. This figure shows to what extent the prediction label predicted by the learning model was correct for the teacher data, that is the true correct data. In the lower right, the cells are shaded in gray, the upper row 38.9% shows the correct answer rate for the entire data set, and the lower row 61.1% shows the wrong answer rate. The squares painted in green indicate the model answered correctly, and the squares painted in orange indicate the incorrect predictions. Of the upper and lower numbers in the green and orange cells, the upper part is the number of classified data in that cell, and the lower part is the ratio of classified data to the total number of data sets. The white areas indicate the percentage of data for each true label and output label. When the test data was predicted to be chalcopyrite and it was actually chalcopyrite, a correct answer rate of 50.0% was obtained. The test data predicted that it was chalcopyrite, but the error rate was 50.0%, indicating that it was actually a different mineral.

| | Galena | Hematite big particle | Hematite small particle | Hematite very small particle | Chalco-pyrite | True labels |
|---|---|---|---|---|---|---|
| Galena | 0<br>0.0% | 0<br>0.0% | 0<br>0.0% | 0<br>0.0% | 0<br>0.0% | NaN%<br>NaN% |
| Hematite big particle | 46<br>9.3% | 31<br>6.3% | 9<br>1.8% | 10<br>2.0% | 5<br>1.0% | 30.7%<br>69.3% |
| Hematite small particle | 3<br>0.6% | 0<br>0.0% | 66<br>13.3% | 0<br>0.0% | 33<br>6.7% | 64.7%<br>35.3% |
| Hematite very small particle | 48<br>9.7% | 63<br>12.7% | 24<br>4.8% | 93<br>18.8% | 59<br>11.9% | 32.4%<br>67.6% |
| Chalco-pyrite | 0<br>0.0% | 1<br>0.2% | 2<br>0.4% | 0<br>0.0% | 3<br>0.6% | 50.0%<br>50.0% |
| **Output labels** | 0.0%<br>100% | 32.6%<br>67.4% | 65.3%<br>34.7% | 90.3%<br>9.7% | 3.0%<br>97.0% | **38.9%**<br>**61.1%** |
| | Galena | Hematite big particle | Hematite small particle | Hematite very small particle | Chalco-pyrite | **True labels** |

**Figure 11.** Confusion matrix for the RGB image deep learning results for the test data.

### 3.2. Deep Learning Analysis for Hyperspectral Data

In Section 3.1, we described the deep learning that was performed on RGB images of five types of minerals, resulting in low-precision classification results. Here, the results of deep learning analysis for hyperspectral data instead of RGB images are shown. For the hyperspectral data, we constructed a neural network consisting of a convolutional layer, normalization layer, pooling layer, and softmax layer from scratch as there was no pre-trained network for hyperspectral data. The CNN for hyperspectral data was created based on a CNN called VGG-16, which was highly evaluated in the 2014 ImageNet Large Scale Visual Recognition Challenge (ILSVRC) [20]. The accuracy of the classification was improved by reducing the size of the filter and increasing the number of layers. The split data set consisted of training data, validation data, and test data. The training data was used for training the network as shown in Table 4, the validation data was used for feedback for the output results of the training, and the test data was used for testing the trained model. In addition, the parameters for deep learning are shown in Table 5. In deep learning, the gradient method was used to learn the data. Among them, ADAM (adaptive moment estimation) was used as an optimizer for hyperspectral data and SGDM (stochastic gradient descent momentum) was used. In addition, the mini-batch size, max epochs, and the calculation time are shown in Table 5 as well.

**Table 4.** Hyperspectral data set.

| Hyperspectral Data $1 \times 1 \times 204$ Pixels | Training Data | Validation Data | Test Data | Total |
|---|---|---|---|---|
| Galena | 770 | 96 | 97 | 963 |
| Chalcopyrite | 770 | 97 | 97 | 994 |
| Hematite large particles | 765 | 95 | 96 | 956 |
| Hematite small particles | 806 | 101 | 101 | 1008 |
| Hematite very small particles | 819 | 103 | 102 | 1024 |

**Table 5.** Learning options of deep learning. ADAM (adaptive moment estimation).

| | Hyperspectral Data | | | RGB Images |
|---|---|---|---|---|
| **Learning Options** | **Two Types of Minerals** | **Three Different Grain Sizes of Hematite** | **Five Types of Minerals** | **Five Types of Minerals** |
| Optimizer | ADAM | ADAM | ADAM | SGDM |
| Mini Batch Size | 100 | 100 | 100 | 100 |
| Max Epochs | 25 | 50 | 100 | 75 |
| Elapsed time | 15 min | 61 min | 253 min | 94 min |
| Initial Learn Rate | 1.00E-04 | 1.00E-04 | 1.00E-04 | 1.00E-04 |

In the deep learning of hyperspectral data, the data set of chalcopyrite and galena was analyzed first to see if the mineral species can be identified, and then deep learning analysis for three hematite species with different grain sizes was performed. Then, as the final step, an experiment was conducted to determine whether or not the five types can be classified using deep learning. The learning curve for the identification of mineral species is shown in Figure 12. The horizontal axis shows the iteration and the vertical axis shows the accuracy of classification. In deep learning, a dataset is divided into a plurality of pieces in order to improve the computational ability, and then learning is repeatedly performed on the divided datasets called mini-batches. The number of iterations indicates how many times learning was repeated for these patches. In Figure 12, the blue line shows the accuracy for learning the data set and the black dots represent the accuracy for the verification data set. In deep learning, when training a network, the training data is used to train the network, and then the verification data is used to verify and feedback to the network to change the weighting.

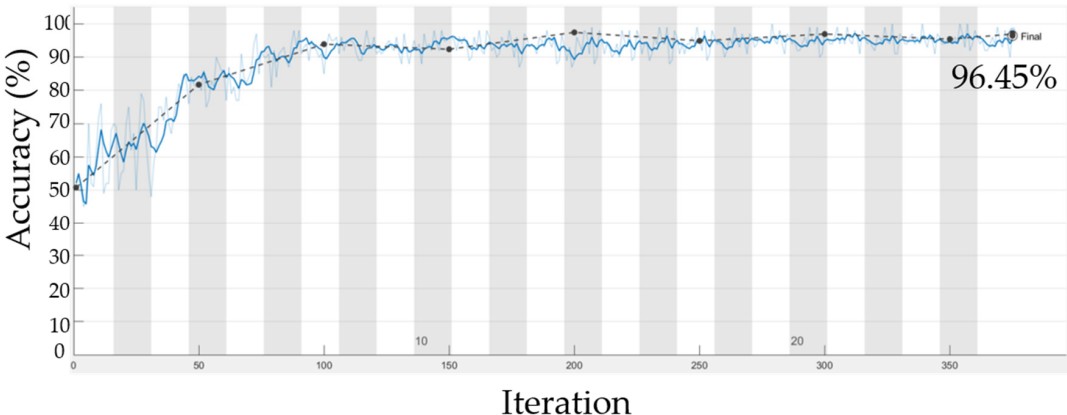

**Figure 12.** The learning curve of deep learning for hyperspectral data. (The horizontal axis shows the iteration and the vertical axis shows the accuracy of classification. The blue line shows the accuracy for the learning data set and the black dots represent the accuracy for the verification data set).

By repeating this, it is possible to build a highly robust network. In Figure 12, the accuracy sharply increased until the number of calculations exceeded 100, and as the number of calculations increased,

the accuracy of classification gradually approached 100% and, finally, reached 96.45%. The learning curve tended to rise in a zigzag manner because the learning accuracy decreased when the learning target moved to a different patch. From these results, we confirmed that, by analyzing the hyperspectral data with deep learning, it was possible to classify mineral types with a high accuracy of 96.45%. In addition, Figure 13 shows the confusion matrix of the learning results of the classification of the two types of minerals, in which the performance of the classifier was evaluated using test data that was not used for learning. This shows a high classification accuracy of over 90% for the test data.

|  | | | |
|---|---|---|---|
| Galena | 96<br>49.2% | 4<br>2.1% | 96.0%<br>4.0% |
| Chalco-<br>pyrite | 0<br>0.0% | 95<br>48.7% | 100.0%<br>0.0% |
| **Output<br>labels** | 100%<br>0.0% | 96.0%<br>4.0% | **97.9%**<br>**2.1%** |
|  | Galena | Chalco-<br>pyrite | **True<br>labels** |

**Figure 13.** Confusion matrix of the deep learning results for two types of rocks using test data.

Next, hyperspectral data was acquired for three types of mineral samples with different grain sizes, and they were similarly classified by deep learning. The learning curve is shown in Figure 14. By analyzing the hyperspectral data of the same mineral with deep learning, it was possible to classify with accuracy as high as 94.31%, similar to the identification of mineral species. Figure 15 shows the confusion matrix of the learning results of the classification of the three different grain sizes of hematite, in which the performance of the classifier was evaluated using test data that was not used for learning. The classification accuracy of the training model for the test data of hematite small particles was 86.1%; however, the overall classification accuracy was 92.3%.

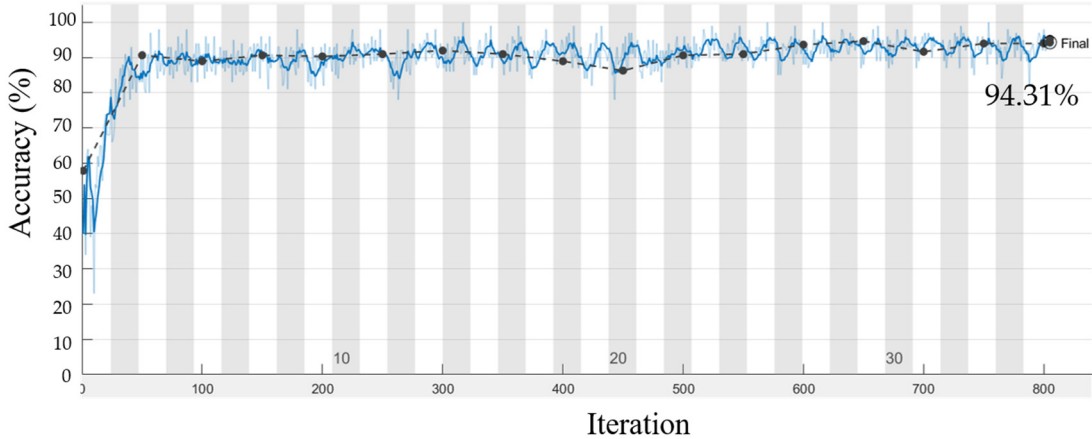

**Figure 14.** The learning curve of deep learning for three types of minerals. (The horizontal axis shows the iteration and the vertical axis shows the accuracy of classification. The blue line shows the accuracy for the learning data set and the black dots represent the accuracy for the verification data set.).

| Hematite big particle | 82 27.4% | 8 2.7% | 0 0.0% | 91.1% 8.9% |
|---|---|---|---|---|
| Hematite small particle | 14 4.7% | 93 31.1% | 1 0.3% | 86.1% 13.9% |
| Hematite very small particle | 0 0.0% | 0 0.0% | 101 33.8% | 100% 0.0% |
| **Output labels** | 85.4% 14.6% | 92.1% 7.9% | 99.0% 1.0% | **92.3% 7.7%** |
| | Hematite big particle | Hematite small particle | Hematite very small particle | **True labels** |

**Figure 15.** Confusion matrix for three different types of grain sizes of test data.

Finally, we tested whether we could identify mineral types and grain sizes using CNN by inputting the data of three different grain sizes of hematite as well as chalcopyrite and galena. The three particle size fractions performed in this study, large, small, and very small, showed that micro-order classification can be achieved through a combination of deep learning and hyperspectral imaging. The versatility and scalability of deep learning provides fundamental suggestions for more practical use in the future with a higher number of classifications. The results of the inverse analysis using Grad-CAM, discussed below, showed that significant changes in spectral shape occur depending on the particle size, which is an important fundamental study for the further refinement of the technology in the future.

As shown in the learning curve diagram in Figure 16, we were able to classify the learning data with a high accuracy of 91.33%. These results show that it is possible to classify mineral species and particle sizes with high accuracy by analyzing hyperspectral data with deep learning. Compared with the low accuracy of mineral identification of 38.9% using RGB images, we considered that the hyperspectral data has wavelength information that uniquely identifies minerals better than RGB images. In addition, Figure 17 shows the confusion matrix of the learning results of the classification of the five types of minerals, in which the performance of the classifier was evaluated using test data that was not used for learning. The overall accuracy of the classification of the five ores was 91.9%.

As mentioned above, the verification was carried out by dividing into three stages whether the difference in mineral species and grain size could be performed by using deep learning on hyperspectral data. From the results, we succeeded in identifying the mineral type and particle size with a high accuracy of 91.9% by analyzing the hyperspectral data of minerals using deep learning. It was possible to non-destructively sort the mineral species and characteristics before the beneficiation process.

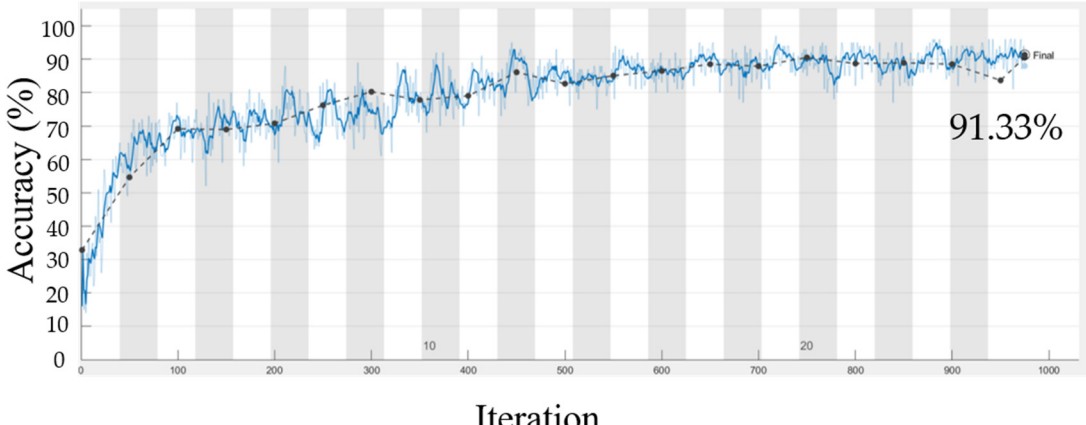

**Figure 16.** The learning curve of deep learning for five types of minerals. (The horizontal axis shows the iteration and the vertical axis shows the accuracy of classification. The blue line shows the accuracy for the learning data set and the black dots represent the accuracy for the verification data set.).

| | Galena | Hematite Small | Hematite Very Small | Hematite big | Chalco-pyrite | True labels |
|---|---|---|---|---|---|---|
| **Galena** | 94<br>19.0% | 4<br>0.8% | 0<br>0.0% | 0<br>0.0% | 1<br>0.2% | 94.9%<br>5.1% |
| **Hematite Small** | 0<br>0.0% | 81<br>16.4% | 13<br>2.6% | 0<br>0.0% | 1<br>0.2% | 85.3%<br>14.7% |
| **Hematite Very Small** | 1<br>0.2% | 9<br>1.8% | 80<br>16.2% | 0<br>0.0% | 0<br>0.0% | 88.9%<br>11.1% |
| **Hematite Big** | 0<br>0.0% | 0<br>0.0% | 0<br>0.0% | 102<br>20.6% | 0<br>0.0% | 100.0%<br>0.0% |
| **Chalco-pyrite** | 1<br>0.2% | 2<br>0.4% | 8<br>1.6% | 0<br>0.0% | 97<br>19.6% | 89.8%<br>10.2% |
| **Output labels** | 97.9%<br>2.1% | 84.4%<br>15.6% | 79.2%<br>20.8% | 100%<br>0.0% | 98.0%<br>2.0% | **91.9%**<br>**8.1%** |

**Figure 17.** Confusion matrix of the deep learning results.

## 4. Discussion

We analyzed the remaining 8.1% of data with the wrong answer and discuss the reasons for the wrong answers. Figure 18 shows the true spectra of the minerals (ground truth data) and the spectrums of the wrong answers (misclassified data). The horizontal axis represents the wavelength and the vertical axis represents the intensity of reflection. In Figure 18a, the spectra correctly classified as galena (solid line) and those misclassified as other than galena (broken line) are indicated. Others are similarly pointed to the correctly classified and misclassified spectra. The shape of the spectrum differs between the ground truth and misclassified data. Figure 19 shows the results of Grad-CAM, an inverse analysis of the CNN, which shows where the CNN focused on in the input data for classification. The vertical axis represents the colormap intensity and the horizontal axis represents the wavelength.

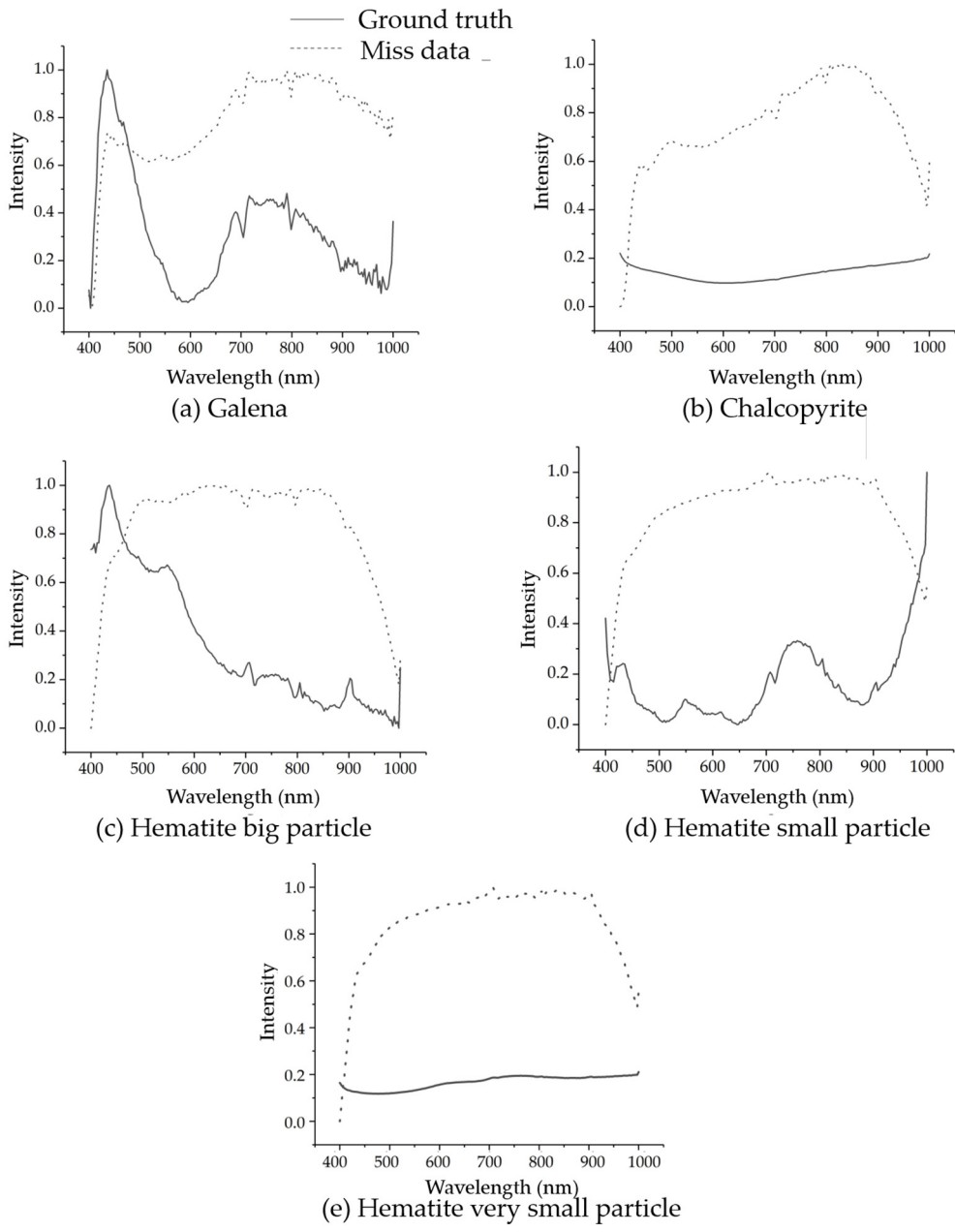

**Figure 18.** Comparison between the ground truth spectrum and misclassified spectrum of (**a**) galena, (**b**) chalcopyrite, (**c**) hematite large particles, (**d**) hematite small particles, and (**e**) hematite very small particles. The solid line shows the ground true spectrum and the broken line shows.

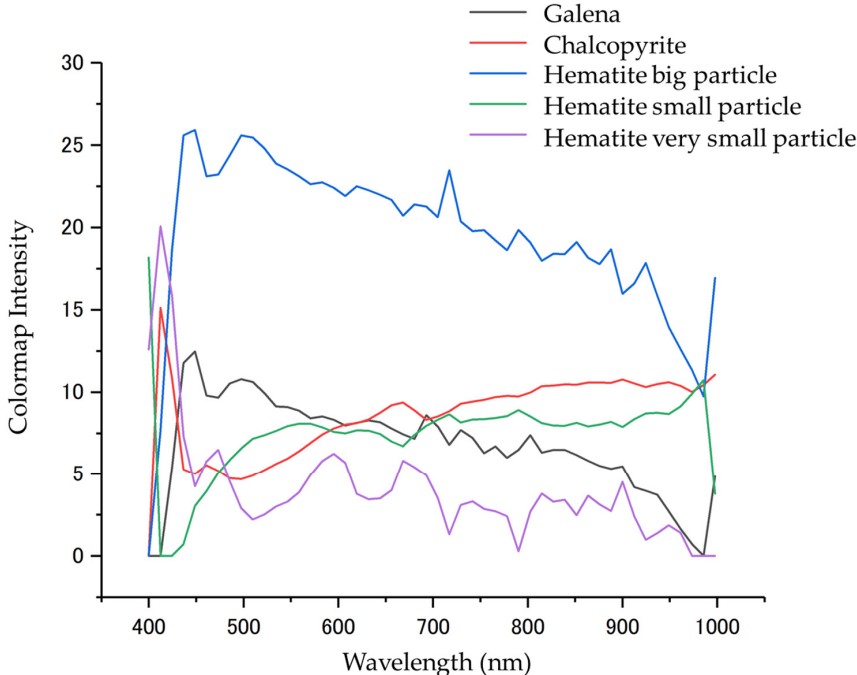

**Figure 19.** Results of the Grad-CAM analysis for the deep learning results.

Although there was a difference in the spectral shape between the ground truth and the misclassified data in Figure 18, the use of Grad-CAM allows us to show which wavelength region of the spectrum was affected by the misclassification. A higher colormap strength on the vertical axis indicates that the influence of this wavelength band was significant in classification, while a lower colormap strength indicates that the data in this band was not important in the classification using CNN.

When comparing both figures, there is an error between the ground truth spectrum and the incorrectly answered spectrum at the wavelengths with high colormap strength in Grad-CAM. From a hyperspectral mineralogical point of view, characteristic spectra of mineral species occur in the high wavelength range, such as in the infrared, however, the results of the Grad-CAM analysis showed that specific spectra that identify minerals appear even in the short-wavelength range. For chalcopyrite, there is a large difference between the ground truth and misclassified data around 450 nm, where Grad-CAM indicates that this was important. For the three hematite types with different grain sizes, there is a large difference in shape between the misclassified data as a whole and the correct data, especially at the two ends of the wavelengths to which Grad-CAM points, which has a significant impact on discrimination; for the three Hematite types, the largest misclassification in the group indicates that the difference between the two ends of the wavelengths was subtle.

As both ends of the hyperspectral data acquired by this system are sensitive from the viewpoint of spectroscopicity, it is difficult to handle them. In the future, the data must be expanded to construct a systematic system. It will be necessary to construct a system that feeds back the information from these erroneous data to the CNN; however, this system alone is sufficiently accurate enough to be used in practical work as it has shown a high classification accuracy of more than 90% for the five types of minerals.

In this paper, we described an identification experiment of mineral species using deep learning and hyperspectral imaging. By using deep learning, a large amount of data can be processed at high speed, and features that determine the mineral species can be automatically extracted and learned from the input data of minerals. When an expert identifies a mineral, they conduct an appraisal by focusing on the color and transparency of the mineral; however, by using deep learning, the feature points to be noted when classifying the mineral are automatically detected. As the naked eye information is a great

clue in expert identification, we performed deep learning on RGB images to identify minerals; however, the classification accuracy was 32.66% and proved to be difficult to identify with only RGB images.

Thus, we considered that an RGB image alone is not sufficient for mineral identification, because the identification by an expert with their naked eyes is performed not only by the color and transparency of the mineral surface, but also by a composite viewpoint, such as the specific gravity and hardness. Therefore, we obtained hyperspectral data with the data from a wider area and with higher resolution than the RGB images and analyzed this using deep learning. As a result, the classification accuracy of mineral species was 91.9%, which was better than that using RGB images. The accuracy of discrimination increased significantly compared with using RGB images. From this, we suggest that the wavelength band that defines the mineral species is obtained outside the wavelength region obtained in the RGB image, and propose that the hyperspectral data includes the characteristic optical properties of the mineral.

## 5. Conclusions

With hyperspectral imaging, high resolution spectral data that contains information from the visible light wavelength region to the near infrared region can be obtained. Using deep learning, the features of the hyperspectral data can be extracted and learned, and the spectral pattern that is unique to each mineral can be identified and analyzed. For our experiment, we prepared five types of minerals, galena, chalcopyrite, hematite with large particles (0.5–1.0 cm), hematite with small particles (<100 μm), and hematite with very small particles (<20 μm) and conducted the classification task using hyperspectral imaging and deep learning.

In the hyperspectral data classification task for mineral types and grain sizes using deep learning, the accuracy was 91.9% providing in high-accuracy classification results. On the other hand, the accuracy was 39.52% for the RGB images resulting in low-precision classification results compared with using hyperspectral data. For the misclassified data, we discussed the results of the inverse analysis of the CNN output using Grad-CAM. The results of the analysis revealed that the misclassification was caused by differences between the ground truth and the misclassified data in a specific wavelength region. By reflecting the results of this analysis in the CNN, a more robust network can be constructed.

This automatic mineral identification system can determine not only the type of mineral but also the size of the crystal of the mineral at the same time. By adopting deep learning, it is possible to process a large amount of data at high speed. Through executing this system on minerals after mining, it is possible to distinguish the types and characteristics of the minerals and select appropriate beneficiation means in the subsequent beneficiation process. From the above, we found that the method combining deep learning and hyperspectral imaging was effective in identifying mineral species and characteristics, and these judgements were possible with high accuracy, high speed, and low costs.

**Author Contributions:** Conceptualization, Y.K.; investigation, N.O. (Natsuo Okada), Y.M., N.O. (Narihiro Owada), and K.H.; writing—original draft preparation, N.O. (Natsuo Okada); writing—review and editing, Y.K. and K.H.; supervision, Y.K. and A.S.; All authors have read and agreed to the published version of the manuscript.

**Funding:** This research was supported by Akita University Support for Fostering Research Project.

**Conflicts of Interest:** The authors declare no conflict of interest.

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
