# Peer review of "Automated Identification of Mineral Types and Grain Size Using Hyperspectral Imaging and Deep Learning for Mineral Processing"

_minerals, doi:10.3390/min10090809_

Round 1

Reviewer 1 Report

The application of deep learning for automated mineralogy analysis is of interested, but the following issues prevent this research from being sufficiently impactful:

  1. Insufficient reference to existing literature on geometallurgy, process mineralogy, and similar attempts to apply data analytics and machine learning to automated mineralogy analysis is cited.
  2. The authors are too vague on how automated mineralogy analysis would improve mineral processing productivity and increase recovery for lower grade deposits. More detail should be provided in how this would be achieved. How exactly would the output of the models (classification on small subsets of the samples) be incorporated to assist improved process productivity?
  3. Insufficient information is provided on the experimental data: How many samples were analysed? How was the ground truth determined? Why were the specific classes of minerals and particle sizes chosen? How do these minerals and particle sizes have industrial relevance?
  4. It is unclear how many samples were used, but it seems that the number of samples are less than 1000 - this is a very small sample size to train a convolutional neural network from scratch!
  5. How was the training, validation and testing data sets constructed? Was there a validation data set? A validation data set is required for hyperparameter optimization, to prevent data snooping of the test set.
  6. What hyperparameters are present in the models (e.g. number of layers)? What are the sensitivity of the results to different values of these hyperparameters? How would the hyperparameters for a new case study be chosen?
  7. The transformation of the hyperspectral data to profiles, instead of using slices as input for the CNN, is not clearly explained or motivated.
  8. The confusion matrix as in figure 12 should be provided for all the models applied, and clearly indicated whether it is for training, validation or testing data.
  9. Provide some examples of correct and missed classifications, and consider using tools such as deep dream to interpret why the missed classifications occur.

Author Response

Dear Reviewr of my thesis.

Regards,

Natsuo okada

Reviewer 2 Report

The topic/idea is very appealing and can attract a certain number of readers. The paper is well written (easy to read). However, it needs a major revision to be considered if it can be accepted or not due to the following reasons:

  • Deep learning in itself is a black-box, the authors put this black-box into another box and presented in here.The data cleaning, processing, size of the input data, model parameters are not discussed in the paper. Please find my comments in the attached pdf file. 
  • Results are analysed in a very shallow way. Scientific argumentation is missing. 
  • The identified an identification gap is practice (real-mine), however, they haven't discussed how their method can be applied in the real applications. 
  • No information on the selection of training, verification and test datasets are given.

The following points can be discussed in the paper:

  • how robust is their model for the identification of other types of minerals (instead of only three mineral types).
  • Do you need to retrain your model? Are there a mechanism to feed back the output of test data analysis to the trained model?

Author Response

(The authors gave the same response as above.)

Round 2

Reviewer 1 Report

This reviewer notes the care taken by the authors to address the comments made on the first draft. Feedback on the original points raised is provided here:

1. Insufficient references to existing literature, geometallurgy, process mineralogy, etc.

Adding four references without providing a synopsis / summary of how geometallurgy / process mineralogy can be implemented in practice is not sufficiently addressing the original comment. A search for "geometallurgy" on Scopus delivers more than 200 references, while "process mineralogy" delivers more than 348 references. The authors should be aware that this is an active research field, with sufficient context and background (and grounding in existing approaches) to be included in the article. See for example Pierre-Henri Koch, Jan Rosenkranz, Sequential decision-making in mining and processing based on geometallurgical inputs, Minerals Engineering, Volume 149, 2020.

2. More details on how automated mineralogy would improve mineral processing productivity

The revision made (addition of figure 1) is not sufficient - figure 1 just indicates that low grade material will be further delayed for future processing. Furthermore, insufficient detail is provided where in the process separation would take place, and how. The incorporation of particle size information in the flow diagram is also unclear. See previous reference for an example of the type of detail / decision-making that could be considered.

3. Particle size motivation (and other comments).

Mineral particle size is of course relevant, but the suggested approach only distinguishes between three pre-determined particle sizes (with the output being a categorical "very small", "smalll or "big" grain) - this is not the same as a quantified particle size distribution with many bins, with each size bin containing a cumulative fraction, as is typically employed for comminution and separation process optimization. The authors need to provide information how their method that is only trained for three specific sizes ("very small" yes or no, "small" yes or no, "big yes or no) can be adapted for practical use with more useful particle size distributions. (The other comments have been addressed; except for the deficiencies of the new figure 1; see previous point)

4. Sample size

Thank you for providing the required information. Note that it is not a fair comparison when the RGB data set is a smaller data set than the hyperspectral data set. The authors need to explain this difference, and contextualize the comparison of the results based on the different sized data sets. (How is one method more likely to perform better due to more training data being available?)

5. Training, validation and test data

Thank you for addressing this comment.

6. Hyperparameters

Thank you for addressing this comment.

7. Hyperspectral slices

Figure 3 still shows hyperspectral slices directly as input to convolution and normalization... Processing of hyperspectral data as suggested in figure 7 is one preprocessing approach, but the motivation provided for using this approach as opposed to what is indicated in figure 3 is not clear.

8. Confusion matrices

Thank you for addressing this comment.

9. Interpretation of missed classification.

Thank you for the use of Grad-CAM to inspect the incorrectly classified examples. Very peculiar (and similar) patterns appear to be present - there may be value to interpret this from a hyperspectral mineralogical perspective.

Please note that the technical English use in the article must be improved.

Author Response

Thank you for your kind editing. We have responded to your comments. Please see the attachment.

Reviewer 2 Report

Dear Authors, 

Many thanks for the responses. The manuscript has been improved a lot. I have only three comments.

  1. Re: Response 18:

    "In actual mine, can you control the intensity of the light as it is required (imagine how many measurements should be done in a day) and also in the reality we don't have always a flat surface. How do you want to overcome this problem?" this is very important, please elaborate. 

  2. Improve your figure and Formula quality (e.g. Figure 4 is not readable). Try to print out for yourself first and check if all the information are readable in A4 for the readers.
  3. Citation must be improved. Please double check to cite all the references that you have used. (e.g. reference is missing) 

Author Response

Thank you for your kind reviewing. We have responded to your comment. Please see the attachment.
